# Maternal HIV and syphilis are not syndemic in Brazil: Hot spot analysis of the two epidemics

**Mary Catherine Cambou**[1,2]*, **Eduardo Saad**[3], **Kaitlyn McBride**[2], **Trevon Fuller**[3], **Emma Swayze**[4], **Karin Nielsen-Saines**[3]

**1** Department of Medicine, Division of Infectious Diseases, UCLA David Geffen School of Medicine, Los Angeles, California, United States of America, **2** Department of Health Policy and Management, UCLA Fielding School of Public Health, Los Angeles, California, United States of America, **3** Department of Pediatrics, Division of Pediatric Infectious Diseases, UCLA David Geffen School of Medicine, Los Angeles, California, United States of America, **4** Department of Medicine, Western Michigan University Homer Stryker School of Medicine, Kalamazoo, Michigan, United States of America

* mcambou@mednet.ucla.edu

**Data Availability Statement:** All data are available by request from the Brazilian Ministry of Health/ SINAN via the e-SIC platform: 1.https://esic.cgu. gov.br/falabr.html?aspxerrorpath=/sistema/site/

## Abstract

While the annual incidence of HIV diagnosis in pregnancy in Brazil remains relatively stable, rates of maternal syphilis increased over six-fold in the past decade. We hypothesized that maternal HIV and syphilis are two distinct epidemics. Data on all cases of maternal HIV or syphilis detected in pregnancy between January 1, 2010 to December 31, 2018 were requested from the Brazilian Ministry of Health. In order to evaluate how the epidemics evolved over the time period, ArcGIS software was used to generate spatiotemporal maps of annual rates of detection of maternal HIV and syphilis in 2010 and 2018. We utilized Euclidean-distance hot spot analysis to identify state-specific clusters in 2010 and 2018. From 2010 to 2018, there were 66,631 cases of maternal HIV, 225,451 cases of maternal syphilis, and 150,414 cases of congenital syphilis in Brazil. The state of Rio Grande do Sul had the highest rate of maternal HIV detection in both 2010 and 2018. Hot spots of maternal HIV were identified in the three most Southern states in both 2010 and 2018 (99% confidence, z-score >2.58, p <0.01). While syphilis incidence >30 per 1,000 live births in 2018 in four states, only the two coastal states of Rio de Janeiro and Espirito Santo in Southeastern Brazil were significant hot spots (90% confidence, z-score 1.65–1.95, p <0.10). Contrary to the general assumption, HIV and syphilis epidemics in Brazil are not syndemic in pregnant women. There is a spatial cluster of maternal HIV in the South, while syphilis is increasing throughout the country, more recently on the coast. Focusing on maternal HIV hot spots in the Southern states is insufficient to curtail the maternal and congenital syphilis epidemics throughout the country. New strategies, including ongoing hot spot analysis, are urgently needed to monitor, identify and treat maternal syphilis.

## Introduction

Mother-to-child transmission (MTCT) of both HIV and syphilis remain significant challenges in Brazil despite increased public health efforts [1–4]. MTCT of HIV is the leading cause of

index.aspx 2.https://falabr.cgu.gov.br/publico/Usuarios/AutoCadastroUsuarioCidadao.aspx The Citizen Information Service (SIC) of the Ministry of Health was established by Ordinance No. 1,583, of 19 July 2012, which refers to the application of the Law on Access to Information within the scope of the Ministry of Health. Information to the Citizen (SIC) has been active since May 2012 and Law No. 12,527 regulates the constitutional guideline the data access. Anyone can register and request data through this platform. The authors did not have special access privileges.

**Funding:** MCC was supported by the UCLA Postdoctoral Fellowship Training Program in Global HIV Prevention Research (PIs: Currier and Gorbach), NIMH grant T32MH080634. EJS was supported by the UCLA South American Program in HIV Prevention Research Program (PI: Clark), NIMH grant R25MH08722. The funders had no role in study design, data collection and analysis, decision to publish, or preparation of the manuscript.

**Competing interests:** The authors have declared that no competing interests exist.

HIV infection among children worldwide. In an effort to curb pediatric HIV infections, the World Health Organization (WHO) released the Option B+ guidelines in 2012, whereby all pregnant women living with HIV would receive lifelong antiretroviral treatment (ART), regardless of CD4+ T lymphocyte count [5]. Brazil is unique in that it offers free, universal ART coverage to all persons living with HIV (PLWH) [6]. As a result, MTCT rates of HIV in Brazil have remained relatively stable over the past ten years. This stands in stark contrast to the syphilis epidemic in Brazil. While maternal syphilis is easily diagnosed in pregnancy, and treatment with penicillin effectively prevents MTCT of syphilis, rates of maternal syphilis in Brazil have increased more than six-fold in the past decade (Fig 1).

Maternal HIV and syphilis are closely linked [7–10]. Syphilis increases the risk of HIV sero-conversion through mucosal disruption and genital inflammation, as evidenced by increased cytokines and T-cell recruitment [11]. In turn, HIV augments the risk of syphilis acquisition, although the mechanism is less clear among pregnant women. HIV and syphilis co-infection increases the likelihood of MTCT, highlighting the need to address both sexually transmitted infections (STI) in public health strategies. In 2010, the Pan American Health Organization (PAHO) committed to the goal of dual elimination of HIV and syphilis MTCT by reducing the HIV MTCT rate to less than 2%, and the congenital syphilis incidence to less than 0.5 cases per 1,000 live births [12]. Cuba was the first country recognized by the WHO for their successful eradication of both HIV and syphilis MTCT [13]. Since 2014, a total of 11 countries received WHO validation for dual elimination of HIV and syphilis MTCT [14]. In response to the PAHO goal of dual elimination, several public policies were implemented in Brazil to increase testing capacity and treatment [15]. Despite these efforts, rates of maternal and congenital syphilis continue to climb in Brazil.

In accordance with the PAHO goal of dual elimination of HIV and syphilis MTCT, efforts to curtail syphilis MTCT in Brazil focus largely on women living with HIV. However, if HIV

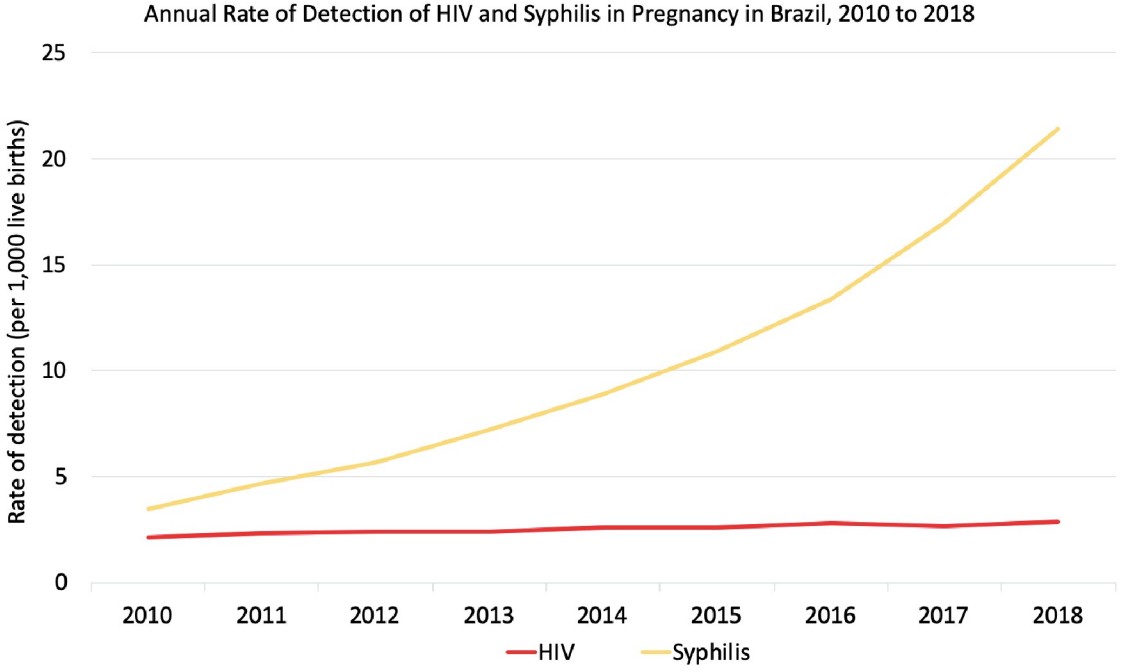

**Fig 1. Annual rate of detection of HIV and syphilis in pregnancy in Brazil, 2010 to 2018 (Modified from the Brazilian Ministry of Health).**

and syphilis are syndemic among pregnant women in Brazil, it is unclear why rates of maternal HIV plateaued, while rates of maternal syphilis increased. We hypothesize that maternal HIV and maternal syphilis cases in Brazil, in fact, reflect two separate epidemics. Geographic information systems (GIS) and hot spot analysis are underutilized research techniques that may provide insight into epidemic patterns [16]. Here, we provide a spatiotemporal analysis and evaluation of hot spots of annual rates of detection of HIV and syphilis in Brazil from 2010 to 2018 [16].

## Materials and methods

Data were provided by the Sistema de Informação de Agravos de Notificação (SINAN), the national reporting system for notifiable diseases of the Brazilian Ministry of Health, for all pregnant women with HIV or syphilis detected between January 1, 2010 to December 31, 2018 [17–19]. All cases of maternal HIV and syphilis are reported to SINAN by law. Data are publicly available if requested, although we used a combination of aggregate and individual-level data provided by the Ministry of Health through the Sistema Eletrônico de Informações ao Cidadão (e-SIC) [20]. The annual rate of detection of maternal HIV (per 1,000 live births) was defined by the Ministry of Health as a new diagnosis during pregnancy (two rapid tests or an ELISA with immunofluorescence/Western Blot), or an established HIV diagnosis. All cases of a reactive treponemal or non-treponemal syphilis test detected during pregnancy are reported to the Ministry of Health. In order to standardize the case definition across the years, and minimize the cases of false-positives or previous infections, we defined the annual rate of detection of maternal syphilis (per 1,000 live births) as a reactive non-treponemal titer (VDRL or RPR) with a confirmatory Treponemal test (fluorescent *Treponemal pallidum* antibody-absorption, micro-hemagglutination *Treponema pallidum* assay, ELISA, or lateral flow) [18]. Each case is reviewed by the Ministry of Health to ensure appropriate diagnosis, adequate maternal, partner and newborn treatment (with a penicillin-based regimen) and compared to previous infections if registered in the system to distinguish between past and present infections. Age at the time of diagnosis was treated as a continuous variable. Self-reported race was categorized as White Black, Mixed/Other or unknown. Education was categorized as illiterate, some primary school, some secondary school, high school graduate or higher, or unknown. Non-treponemal syphilis titers were operationalized as <1:8, ≥1:8 & <1:16, ≥1:16 & <1:128, ≥1:128, and unknown. Syphilis treatment was categorized as penicillin (first-line), non-penicillin, no treatment, or unknown. HIV viral load, CD4+ T lymphocyte count, date of HIV diagnosis and ART regimen were not available. Missing data were treated as missing at random, and reported in Table 1 if applicable.

The spatial unit of analysis was the federative unit of Brazil: the country is divided into 26 states and the Federal District, for a total of 27 federative units. In order to evaluate how the epidemics evolved over the time period, we populated spatiotemporal maps and analyzed hot spots in 2010 and 2018. We used ArcGIS to map the state-specific, annual rates of detection of maternal HIV or syphilis in 2010 and 2018 against a base map of vector data of Brazil. Polygon shapefiles were retrieved from OpenStreetMap [21], an open database that sources from the Instituto Brasileiro de Geografia e Estatística (IBGE), a government-sponsored, open access database [22]. Data are publicly available under the Open Database License [21, 23].

The Getis-Ord Gi* statistic was used to analyze Euclidean-distance hot spot analysis, which has been previously utilized to identify HIV and other STI hot spots [24, 25]. The GI* statistic analyzes whether the rate of detection of the areal unit is higher than expected. The sum of a unit and its neighbors is compared proportionally to the sum of all units using the following

**Table 1. Descriptive characteristics of women with maternal HIV and syphilis detected during pregnancy in Brazil from 2010 to 2018.**

| | HIV[1] | Syphilis[2] |
| --- | --- | --- |
| | N = 66, 631 | N = 225,451 |
| | N (%) | N (%) |
| **Age** | | |
| Median [IQR] | 26 [21, 31] | 23 [19, 28] |
| <20 | 11,357 (17.0) | 61,082 (27.1) |
| 20–29 | 34,327 (51.5) | 117,734 (52.2) |
| ≥30 | 20,947 (31.4) | 46,630 (20.7) |
| Unknown | 1 (<0.01) | 5 (<0.01) |
| **Race** | | |
| White | 24,587 (36.9) | 69,103 (30.6) |
| Black | 9,368 (14.1) | 26,604 (11.8) |
| Mixed/Other | 29,143 (43.7) | 110,911 (49.2) |
| Unknown | 3,534 (5.3) | 18,833 (8.4) |
| **Education** | | |
| Illiterate | 536 (0.8) | 1,786 (0.8) |
| Some Primary School | 23,108 (34.7) | 66,271 (29.4) |
| Completed Primary or Some Secondary School | 15,374 (23.1) | 52,156 (23.1) |
| High School Graduate or Higher | 14,187 (21.3) | 41,181 (18.3) |
| Unknown | 13,427 (20.1) | 64,057 (28.4) |
| **Syphilis Titer** | N/A | |
| <1:8 | | 89,261 (39.6) |
| ≥1:8 & <1:16 | | 33,510 (14.9) |
| ≥1:16 & <1:128 | | 80,424 (35.7) |
| ≥1:128 | | 15,232 (6.7) |
| Unknown | | 7,024 (3.1) |
| **Syphilis Treatment** | N/A | |
| Penicillin | | 197,218 (87.5) |
| Non-Penicillin | | 5,797 (2.6) |
| No Treatment Received | | 12,990 (5.7) |
| Unknown | | 9,446 (4.2) |

1) Maternal HIV was defined as a new diagnosis during pregnancy (two rapid tests or ELISA + immunofluorescence/ Western Blot) or established diagnosis

2) Maternal syphilis was defined as a reactive VDRL or RPR titer with a confirmatory *Treponemal* test (fluorescent *Treponemal pallidum* antibody-absorption, micro-hemagglutination *Treponema pallidum* assay, ELISA or lateral flow)

N/A = not applicable

equation [24]:

$$G_i^* = \frac{\sum_{j=1}^{n} w_{i,j} x_j - \bar{X} \sum_{j=1}^{n} w_{i,j}}{s \sqrt{\frac{n \sum_{j=1}^{n} w_{i,j}^2 - \left(\sum_{j=1}^{n} w_{i,j}\right)^2}{n-1}}}$$

Variable $x_j$ represents the maternal HIV or syphilis rate of detection for state j, $w_{ij}$ represents the spatial weight between states I and j, n represents the total number of areal units (26 states

+ federal district = 27 total), and $s = \sqrt{\frac{\sum_{j=1}^{n} x_j^2}{n} - \bar{x}}$. A local sum that is higher than the expected sum, and thus unlikely to be due to random chance alone, results in a statistically-significant z-score [26]. Hot spots represent significant clusters of high z-score values, whereas cold spots represent clusters of low z-score values. Therefore, hot spot analysis is distinct from a simple heat map, where a zone may have a high rate of detection by chance alone. We used Global Moran's I for geospatial autocorrelation given inconsistent spatial patterns across the study area [27]. We used STATA Version 16 for descriptive analyses, and ArcGIS Version 10.7 for the maps and hot spot analysis. For hot spots, confidence levels were defined as follows: 99% confidence (z-score >2.58, p <0.01), 95% confidence (z-score 1.96–2.58, p <0.05), and 90% confidence (z-score 1.65–1.95, p <0.10). For cold spots, confidence levels were defined as follows: 99% confidence (z-score <-2.58, p <0.01), 95% confidence (z-score -2.58 –-1.96, p <0.05), and 90% confidence (z-score -1.95 –-1.65, p <0.10). This study used publicly available de-identified data; therefore, the study was IRB exempt.

## Results

From 2010 to 2018, there were 66,631 cases of maternal HIV detected during pregnancy (Table 1). The median age was 26 years, 43.7% of women identified as mixed race, and 21.3% reported a high school education or higher. During the same time period, there were 225,451 cases of maternal syphilis, and 150,414 neonates were treated for presumed congenital syphilis. The median age of maternal syphilis was slightly lower compared to maternal HIV at 23 years, 49.2% identified as mixed race, and 18.3% reported a high school education or higher. Over one-third (39.6%) had non-treponemal titers <1:8, and 87.5% received first-line penicillin treatment.

Fig 2A & 2B show the descriptive GIS maps of detection of maternal HIV and syphilis in 2010, respectively. In 2010, maternal HIV rates ranged from 0.6 per 1,000 live births to 7.3 per 1,000 live births in the state of Rio Grande do Sul. Maternal syphilis rates ranged from 1.5 to 13.2 per 1,000 live births. Fig 2C & 2D show state-specific hot spot analyses of maternal HIV and syphilis in 2010. Hot spots of maternal HIV were identified in the three most Southern states: Parana, Santa Catarina, and Rio Grande do Sul (99% confidence). Piauí was identified as a maternal HIV cold spot (95% confidence). Hot spots of maternal syphilis were identified in Amapá, and Pará (95% confidence). Mato Grosso was also identified as a maternal syphilis hot spot, although at a confidence level of 90%.

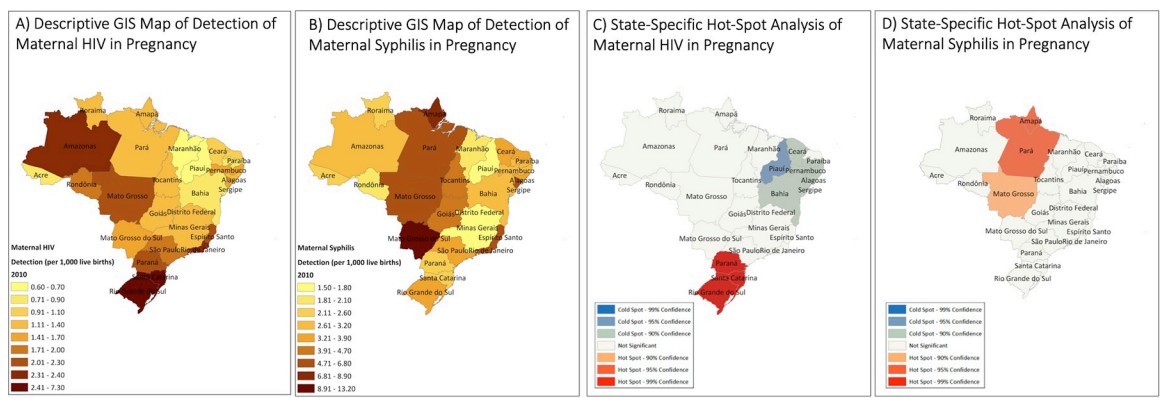

Polygon shapefiles were retrieved from OpenStreetMap [21], an open database that sources from the Instituto Brasileiro de Geografia e Estatística (IBGE) [22], a government-sponsored, open access database. Data are publicly available under the Open Database License [21, 23].

**Fig 2. Spatiotemporal maps of the annual rate of detection of maternal HIV and syphilis among pregnant women in Brazil, 2010.**

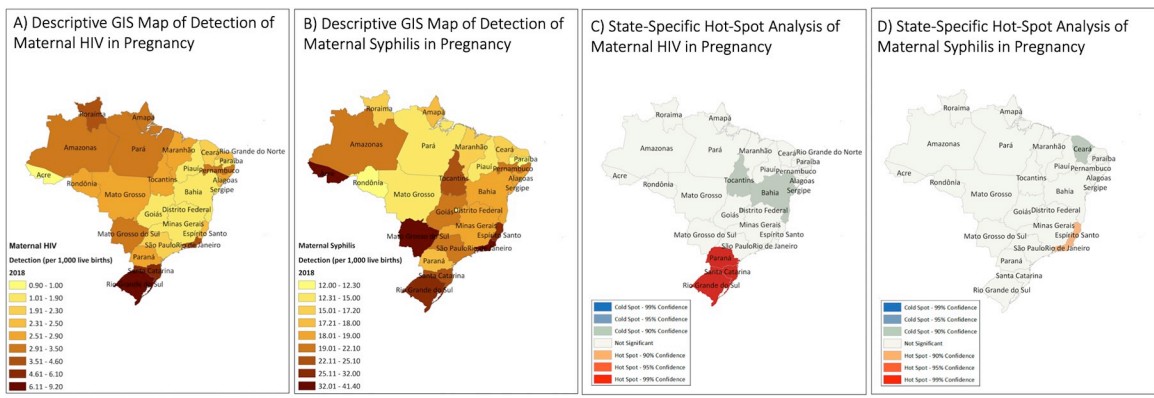

Polygon shapefiles were retrieved from OpenStreetMap [21], an open database that sources from the Instituto Brasileiro de Geografia e Estatística (IBGE) [22], a government-sponsored, open access database. Data are publicly available under the Open Database License [21, 23].

**Fig 3. Spatiotemporal maps of the annual rate of detection of maternal HIV and syphilis among pregnant women in Brazil, 2018.**

Fig 3A & 3B show the descriptive GIS maps of detection of maternal HIV and syphilis in 2018, respectively. In 2018, the highest maternal HIV rate was again in Rio Grande do Sul (9.2 per 1,000 live births). Maternal syphilis rates ranged from 12.0 to 41.4 per 1,000 live births in 2018. The rate of detection of maternal syphilis exceeded 30 per 1,000 live births in four states: Acre, Mato Grosso do Sul, Rio de Janeiro, and Espirito Santo. Fig 3C & 3D show state-specific hot spot analyses of maternal HIV and syphilis in 2018. Hot spots of maternal HIV were again identified in the three most Southern states (99% confidence), the same pattern observed in 2010. The coastal states of Rio de Janeiro and Espirito Santo were the only maternal syphilis hot spots, although at a confidence level of 90%.

## Discussion

Hot spot analysis of maternal HIV and syphilis in Brazil suggests that the two epidemics are distinct: there is a spatial cluster of maternal HIV in the South, while maternal syphilis is on the rise throughout the country, more recently on the Southeastern coast. While HIV and syphilis in pregnancy are intertwined, the current approach in Brazil may neglect management of maternal syphilis as a separate epidemic that requires its own dedicated, individualized public health approach.

The Unified Health System (Sistema Único de Saúde, SUS) HIV treatment program in Brazil is potentially one of the most comprehensive HIV public health programs worldwide, having been tremendously successful over the years [28]. The stable annual HIV MTCT rate over the past decade serves as a testament to the effective implementation of Option B+ guidelines, and universal ART coverage for all PLWH. In addition, the rise of raltegravir-based ART regimens may contribute to sustained viral load suppression among pregnant women, and prevention of HIV MTCT as a result in Brazil [29]. While efforts to curtail the maternal and congenital syphilis epidemics in Brazil have leveraged the existing HIV healthcare infrastructure, Brazil boasts a diverse population of over 200 million, and may require more tailored, region-specific strategies.

The increasing rates of maternal syphilis over the past decade are likely multi-factorial. Increased syphilis testing and enhanced surveillance as part of the *Rede Cegonha* and *Projeto Nascer* campaigns, public policy initiatives aimed at improving prenatal care in Brazil, are possible explanations for increased identification of maternal and congenital syphilis cases throughout the country [15]. While prenatal care is accessible to all pregnant women under

the SUS network in Brazil, the quality may be suboptimal, particularly among women of a lower socioeconomic status [30]. In addition, other studies using spatiotemporal analyses have documented comparable increases in Brazil that cannot be explained by increased testing capacity alone [31, 32]. Inadequate treatment of partners, leading to re-infection of their pregnant partners, and lack of recognition of low-level titer infections (<1:16) without appropriate treatment despite the guidelines [33], have also been cited as possible explanations for rising maternal syphilis rates despite engagement in pre-natal care [3]. A recent study by Swayze et al. found that in a subset of women who fulfilled criteria for a syphilis diagnosis but did not receive penicillin treatment during pregnancy, 83% had been engaged in pre-natal care. This finding challenges the popular narrative that absence of pre-natal care fuels the maternal and congenital syphilis epidemics [3]. This may place unfounded blame on pregnant women and raises questions about this proposed causal relationship.

The Ministry of Health recommends syphilis testing in pregnancy at the first prenatal care visit, during the third trimester (28 weeks), and at delivery [33]. In the public health system, primary care centers are primarily responsible for syphilis testing during prenatal visits, while maternal hospitals test at delivery. Our findings underscore the need to ensure that mandated syphilis testing in pregnancy is implemented in practice. Further, improved communication between institutions within the treatment cascade may increase detection and treatment. The broad adoption of rapid, point-of-care (POC) testing may also contribute to efforts to control syphilis MTCT [34, 35]. While rapid POC syphilis testing is encouraged in the prenatal setting [33], VDRL is still commonly used throughout the country [36–38]. A recent qualitative study on the implementation of point-of-care testing in indigenous communities in the Brazilian Amazon found that 86.7% of individuals diagnosed with syphilis with rapid assays over a 3-month time period were treated. However, significant barriers to care remained, including long travel times and limited access to transportation. The social and economic inequities that shape the maternal syphilis epidemic in Brazil should be addressed in the development of public health strategies to target syphilis MTCT.

This study has limitations. First, we had limited access to maternal HIV data, including the date of diagnosis, HIV viral load, CD4+ T lymphocyte count and ART regimens. We also could not identify the cases of dual HIV/ syphilis infection in the series. However, this should not impact the primary objective of our study. Second, in an effort to capture all cases of maternal syphilis, in 2017, the Ministry of Health defined a case by any positive treponemal or non-treponemal test, regardless of confirmation, in pre-natal care and delivery. In an effort to standardize the case definition across the study years, we defined a maternal syphilis case as a reactive non-treponemal test with a reactive treponemal test. While the increase in 2017 and 2018 must be interpreted with caution, the rising maternal and congenital syphilis cases over the past decade suggest that this change in reporting likely did not substantially impact the linear trend. Third, Euclidean distance hot spot analysis may underestimate variance with <30 areal units of analysis, leading to a Type I error [26]. We cannot link HIV and syphilis databases to examine rates of co-infection, which is a potentially rich area for future research.

## Conclusions

In conclusion, the HIV and syphilis epidemics in pregnancy do not appear syndemic in Brazil. Focusing on maternal HIV hot spots in the Southern states is insufficient to curtail the maternal and congenital syphilis epidemics throughout the country. In order to meet the PAHO goal of less than 0.5 syphilis cases per 1,000 live births, new strategies are urgently needed to identify and treat pregnant women, as well as their partners. Ongoing geographic monitoring allows for improved targeted efforts to tackle dual elimination of MTCT syphilis and HIV in Brazil [16].

## Author Contributions

**Conceptualization:** Mary Catherine Cambou, Karin Nielsen-Saines.

**Data curation:** Mary Catherine Cambou, Eduardo Saad.

**Formal analysis:** Mary Catherine Cambou, Kaitlyn McBride, Trevon Fuller.

**Methodology:** Mary Catherine Cambou, Trevon Fuller.

**Supervision:** Karin Nielsen-Saines.

**Visualization:** Kaitlyn McBride, Trevon Fuller, Emma Swayze.

**Writing – original draft:** Mary Catherine Cambou, Emma Swayze, Karin Nielsen-Saines.

**Writing – review & editing:** Mary Catherine Cambou, Eduardo Saad, Kaitlyn McBride, Trevon Fuller, Karin Nielsen-Saines.

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
