## [Decision Letter · Decision Letter 0]

14 May 2021

PONE-D-21-12741

Maternal HIV and Syphilis are Not Syndemic in Brazil: Hot Spot Analysis of the Two Epidemics

PLOS ONE

Dear Dr. Cambou,

Thank you for submitting your manuscript to PLOS ONE. After careful consideration, we feel that it has merit but does not fully meet PLOS ONE’s publication criteria as it currently stands. Therefore, we invite you to submit a revised version of the manuscript that addresses the points raised during the review process.

We look forward to receiving your revised manuscript.

Kind regards,

Jodie Dionne-Odom, MD

Academic Editor

PLOS ONE

Journal Requirements:

3. We noticed you have some minor occurrence of overlapping text with the following previous publication, which needs to be addressed:

- https://onlinelibrary.wiley.com/doi/full/10.1002/jia2.25547

The text that needs to be addressed involves the Abstract.

In your revision ensure you cite all your sources (including your own works), and quote or rephrase any duplicated text outside the methods section.

Further consideration is dependent on these concerns being addressed.

4. We note that Figures 2 and 3 in your submission contain map images which may be copyrighted.

a. You may seek permission from the original copyright holder of Figures 2 and 3 to publish the content specifically under the CC BY 4.0 license. 

Reviewers' comments:

Reviewer's Responses to Questions

**Comments to the Author**

1. Is the manuscript technically sound, and do the data support the conclusions?

Reviewer #1: Partly

Reviewer #2: Yes

Reviewer #3: Yes

2. Has the statistical analysis been performed appropriately and rigorously? 

Reviewer #1: Yes

Reviewer #2: Yes

Reviewer #3: Yes

3. Have the authors made all data underlying the findings in their manuscript fully available?

Reviewer #1: Yes

Reviewer #2: Yes

Reviewer #3: Yes

4. Is the manuscript presented in an intelligible fashion and written in standard English?

Reviewer #1: Yes

Reviewer #2: Yes

Reviewer #3: Yes

5. Review Comments to the Author

Reviewer #1: The paper entitled “Maternal HIV and Syphilis are not syndemic in Brazil: Hot spot analysis of the two epidemics” approaches an interesting topic but the subject could be better explored by a deeper discussion of data. Below, I describe some comments and suggestions.

Introduction:

Line 10 - Figure 1 is not needed.

Methods:

Line 34 – it was not clear if what data the authors used. Did they asked for nominal data or did they used the one available at the Ministry of Health website? I think it was the second option, it will be good to include the website in the methods section: http://www.aids.gov.br/pt-br/gestores/painel-de-indicadores-epidemiologicos.

Line 38: The authors describe the case definition of syphilis in pregnant women as “Reactive non-treponemal titer with a confirmatory treponemal test”. In 2017, trying to reduce the underreporting of syphilis cases in pregnant women, the Brazilian Ministry of Health changed the cases definition to consider all cases of women diagnosed with syphilis during prenatal care, delivery and/or puerperium should be reported as syphilis in pregnant women and not considered as syphilis in adults how they used to be considered. Also, to have a better control of syphilis cases in pregnancy and avoiding new cases of congenital syphilis, it was considered as a case any test for syphilis in pregnancy (treponemic or non-treponemic). Confirmatory tests are not required to notify a pregnant woman with syphilis. These measures were justified for controlling Congenital syphilis, but they increased the notifications of pregnant women, including women with history of syphilis in the past. So, a careful analysis of the notification data is necessary. It is important to make these considerations in the discussion of the article.

Discussion:

This is the section where the authors should take advantage of data. Brazil is a continental country that presents a lot of diversity. The authors suggest managing syphilis as a separate pandemic, what it is, but it should take advantage of the organized care provided for HIV to implementing its actions. Although prenatal care is easy to access in Brazil and high access rates have been described, the great challenge is the quality of the offered care. It is common to offer rapid test for HIV in prenatal care and offer VDRL test for syphilis (even if the rapid test for syphilis is available).

The authors should discuss these points to give us a better idea of the whole picture. I missed the issues related to HIV. There are many questions that could have been explored to compare the medical care of the two infections in the country.

In addition to the study with indigenous people mentioned in the discussion, there are other important references that discuss the situation of rapid tests in Brazil. The indigenous population is very peculiar and does not represent the national context. Authors should include other data on it.

Reviewer #2: The manuscript aimed at spatiotemporal analysis and evaluation of hotspots of annual rates of detection of HIV and syphilis in Brazil from 2010 to 2018.

The manuscript is well written and the study's outline is well outlined. However, authors should note a few points before publication:

a) Exchange the expression CD4 for CD4 + T lymphocytes.

b) Were there any exclusion criteria in the methodology? For example, notification forms with no information?

c) In the legend of table 1, put the meaning of N / A.

d) Authors should include in the discussion a topic commenting on the possible causes of stability in the diagnosis of HIV infection.

Reviewer #3: This is a very well-written manuscript describing a compelling analysis of maternal HIV and syphilis epidemics in Brazil from 2010 to 2018. The authors found these epidemics to have some geographic distinction, and thus their conclusion—that the maternal syphilis epidemic requires a new, individualized approach to management—seems logical, appropriate, and important.

I have a few questions/comments:

1. Could the authors please further explain the process for detection of maternal syphilis? Does the definition (reactive treponemal + reactive non-treponemal) differentiate incident or prevalent infection from old, previously treated infection? The authors state that the Ministry of Health reviews each case, but what is included in this review (appropriate diagnosis? treatment? etc.)?

2. Lines 112-113: The phrase “low non-treponemal titer infections” is unclear; consider rewording as “lack of recognition of low-level titer infections.” Additionally, the study cited here reports that these low-level titers are associated with lower likelihood of penicillin treatment but does not report that they are an explanation for rising syphilis rates. Perhaps the reference citation for this sentence is incorrect; I’m not sure that inadequate treatment of partners as a cause of rising maternal syphilis infections is discussed in this reference at all.

3. Please double-check if the percentage is 92 (rather than 83%) for cases of maternal syphilis in this study who were engaged in prenatal care.

One additional minor comment:

1. Data is/was should be changed to are/were throughout.

6. PLOS authors have the option to publish the peer review history of their article (what does this mean?). If published, this will include your full peer review and any attached files.

Reviewer #1: No

Reviewer #2: **Yes: **Luiz Fernando Almeida Machado

Reviewer #3: No

---

## [Author Response · Author response to Decision Letter 0]

2 Jul 2021

Dear Dr. Dionne-Odom and esteemed reviewers, 

We thank you for the opportunity to address your edits. We have addressed each point, and feel that the paper is stronger as a result. Thank you again for your consideration. 

Requirements: 

 We apologize about the previous formatting. The manuscript now meets PLOS ONE’s style requirements as outlined by the guidelines. 

 We reviewed each reference individually. The first two references, by Lee et al. and Mukumbang et al, in the original manuscript were cited in error. We have replaced these with the following citations: 

Adachi K, Xu J, Yeganeh N, Camarca M, Morgado MG, Watts DH, et al. Combined evaluation of sexually transmitted infections in HIV-infected pregnant women and infant HIV transmission. PLoS One. 2018;13(1):e0189851. Epub 2018/01/06. doi: 10.1371/journal.pone.0189851. PubMed PMID: 29304083; PubMed Central PMCID: PMCPMC5755782.

João EC, Morrison RL, Shapiro DE, Chakhtoura N, Gouvèa MIS, de Lourdes BTM, et al. Raltegravir versus efavirenz in antiretroviral-naive pregnant women living with HIV (NICHD P1081): an open-label, randomised, controlled, phase 4 trial. Lancet HIV. 2020;7(5):e322-e31. Epub 2020/05/11. doi: 10.1016/s2352-3018(20)30038-2. PubMed PMID: 32386720; PubMed Central PMCID: PMCPMC7323582.

We apologize about this oversight. In addition, we added the following citations per the suggestion by reviewer #1 regarding data access from the Ministry of Health, and per the copyright guidelines regarding basemap citations (please see point 4 for further details): 

Sistema Eletrônico de Informações ao Cidadão (e-SIC) [June 10 2021]. Available from: https://esic.cgu.gov.br/falabr.html

OpenStreetMap: Copyright and License [June 10 2021]. Available from: https://www.openstreetmap.org/copyright.

Instituto Brasileiro de Geografía e Estatística: IBGE [June 10 2021]. Available from: https://www.ibge.gov.br/en/institutional/the-ibge.html.

Open Data Commons Open Database License (ODbL) [June 10 2021 ]. Available from: https://opendatacommons.org/licenses/odbl/.

 Per suggestions from reviewer #1, we added the following citations to strengthen the Discussion: 

Nunn AS, da Fonseca EM, Bastos FI, Gruskin S. AIDS treatment in Brazil: impacts and challenges. Health Aff (Millwood). 2009;28(4):1103-13. Epub 2009/07/15. doi: 10.1377/hlthaff.28.4.1103. PubMed PMID: 19597210; PubMed Central PMCID: PMCPMC2782963.

Pascom ARP, Fonseca FF, Pinho RGG, Perini FB, Pereira G, Avelino-Silva VI. Impact of antiretroviral regimen on viral suppression among pregnant women living with HIV in Brazil. Int J STD AIDS. 2020;31(9):903-10. Epub 2020/07/24. doi: 10.1177/0956462420932688. PubMed PMID: 32702281.

Mario DN, Rigo L, Boclin KLS, Malvestio LMM, Anziliero D, Horta BL, et al. Quality of Prenatal Care in Brazil: National Health Research 2013. Cien Saude Colet. 2019;24(3):1223-32. Epub 2019/03/21. doi: 10.1590/1413-81232018243.13122017. PubMed PMID: 30892541.

Cerda R, Perez F, Domingues RM, Luz PM, Grinsztejn B, Veloso VG, et al. Prenatal Transmission of Syphilis and Human Immunodeficiency Virus in Brazil: Achieving Regional Targets for Elimination. Open Forum Infect Dis. 2015;2(2):ofv073. Epub 2015/07/17. doi: 10.1093/ofid/ofv073. PubMed PMID: 26180825; PubMed Central PMCID: PMCPMC4498254.

Luu M, Ham C, Kamb ML, Caffe S, Hoover KW, Perez F. Syphilis testing in antenatal care: Policies and practices among laboratories in the Americas. Int J Gynaecol Obstet. 2015;130 Suppl 1(Suppl 1):S37-42. Epub 2015/05/17. doi: 10.1016/j.ijgo.2015.04.011. PubMed PMID: 25979116; PubMed Central PMCID: PMCPMC6756481.

Trinh TT, Kamb ML, Luu M, Ham DC, Perez F. Syphilis testing practices in the Americas. Trop Med Int Health. 2017;22(9):1196-203. Epub 2017/06/28. doi: 10.1111/tmi.12920. PubMed PMID: 28653418; PubMed Central PMCID: PMCPMC6764591

Lastly, a reference by Sanchez et al was available on Google scholar but not PubMed, since it was not a medical journal. To facilitate access to all references for our readers, we replaced the citation with the following from ESRI, ArcGIS Pro:

ESRI: ArcGIS Pro. How Hot Spot Analysis (Getis-Ord Gi*) works [June 10 2021]. Available from: https://pro.arcgis.com/en/pro-app/latest/tool-reference/spatial-statistics/h-how-hot-spot-analysis-getis-ord-gi-spatial-stati.htm.

The updated reference list is complete and in accordance with the PLOS ONE references guidelines. We did not have any retracted articles, but feel that the current reference list is stronger because of input from our reviewers. 

3. We noticed you have some minor occurrence of overlapping text with the following previous publication, which needs to be addressed: https://onlinelibrary.wiley.com/doi/full/10.1002/jia2.25547

The text that needs to be addressed involves the Abstract.

In your revision ensure you cite all your sources (including your own works), and quote or rephrase any duplicated text outside the methods section.

Further consideration is dependent on these concerns being addressed.

Thank you for bringing this to our attention. We presented our preliminary findings as a poster at the 23rd International AIDS Society (IAS) Virtual Conference in 2020 (Cambou MC, et al. Are HIV and syphilis syndemic in pregnant women in Brazil? Hot-spot analysis of the two epidemics. Poster PDC0404 presented at: IAS; 2020 July 6-10; Virtual conference). The overlapping text is from the abstract for the poster. However, our findings were not published, and are not under consideration for publication elsewhere. While we cannot cite the abstract, we have added a citation to the poster in the manuscript. In addition, we have modified the abstract slightly to reduce overlapping text with our previous poster abstract.

4. We note that Figures 2 and 3 in your submission contain map images which may be copyrighted.

We appreciate your concern. Figures 2 and 3 were generated in ArcGIS Version 10.7 using publicly available, noncopyrighted polygon shapefiles from OpenStreetMap, an open database that sources from the government-sponsored, open database Instituto Brasileiro de Geografia e Estatística (IBGE). Per the openstreetmap.org/copyright page: 

“OpenStreetMap® is open data, licensed under the Open Data Commons Open Database License (ODbL) by the OpenStreetMap Foundation (OSMF).

You are free to copy, distribute, transmit and adapt our data, as long as you credit OpenStreetMap and its contributors. If you alter or build upon our data, you may distribute the result only under the same license. The full legal code explains your rights and responsibilities.”

 In accordance with the OpenStreetMap guidelines, we have credited them and IBGE in the methods and figures as follows: 

“Polygon shapefiles were retrieved from OpenStreetMap [21], an open database that sources from the Instituto Brasileiro de Geografia e Estatística (IBGE) [22], a government-sponsored, open access database. Data are publicly available under the Open Database License [21, 23].”

Reviewer #1: The paper entitled “Maternal HIV and Syphilis are not syndemic in Brazil: Hot spot analysis of the two epidemics” approaches an interesting topic but the subject could be better explored by a deeper discussion of data. Below, I describe some comments and suggestions.

Introduction:

Line 10 - Figure 1 is not needed.

We appreciate this comment from reviewer #1, who is clearly an expert in maternal HIV and syphilis in Brazil. However, we respectfully request to keep the figure, since most readers are not as familiar with the pattern of maternal HIV and syphilis as the reviewer. 

Methods:

Line 34 – it was not clear if what data the authors used. Did they asked for nominal data or did they used the one available at the Ministry of Health website? I think it was the second option, it will be good to include the website in the methods section: http://www.aids.gov.br/pt-br/gestores/painel-de-indicadores-epidemiologicos.

 We apologize that we did not make this clearer. We used a combination of data sources, including the website you list above, as well as individual-level data requested from the Ministry of Health via e-SIC. Our methods now reflect these distinctions: 

“Data were provided by the Sistema de Informação de Agravos de Notificação (SINAN), the national reporting system for notifiable diseases of the Brazilian Ministry of Health, for all pregnant women with HIV or syphilis detected between January 1, 2010 to December 31, 2018 [17-19]. All cases of maternal HIV and syphilis are reported to SINAN by law. Data are publicly available if requested, although we used a combination of aggregate and individual-level data provided by the Ministry of Health through the Sistema Eletrônico de Informações ao Cidadão (e-SIC) [20].”

Line 38: The authors describe the case definition of syphilis in pregnant women as “Reactive non-treponemal titer with a confirmatory treponemal test”. In 2017, trying to reduce the underreporting of syphilis cases in pregnant women, the Brazilian Ministry of Health changed the cases definition to consider all cases of women diagnosed with syphilis during prenatal care, delivery and/or puerperium should be reported as syphilis in pregnant women and not considered as syphilis in adults how they used to be considered. Also, to have a better control of syphilis cases in pregnancy and avoiding new cases of congenital syphilis, it was considered as a case any test for syphilis in pregnancy (treponemic or non-treponemic). Confirmatory tests are not required to notify a pregnant woman with syphilis. These measures were justified for controlling Congenital syphilis, but they increased the notifications of pregnant women, including women with history of syphilis in the past. So, a careful analysis of the notification data is necessary. It is important to make these considerations in the discussion of the article.

Thank you for this thoughtful comment. We agree that a confirmatory test is not required to notify a pregnant woman with syphilis. However, in order to standardize the definition across the years, we coded a positive maternal syphilis case as a reactive non-treponemal titer with a confirmatory treponemal test, particularly to remove any cases of false-positives or previous infections. Since we had access to individual-level data through e-SIC, we had this data available for all maternal syphilis cases from 2010 to 2018. However, we have now addressed your concern in the limitations section: 

“This study has limitations. First, we had limited access to maternal HIV data, including the date of diagnosis, HIV viral load, CD4+ T lymphocyte count and ART regimens. We also could not identify the cases of dual HIV/ syphilis infection in the series. However, this should not impact the primary objective of our study. Second, in an effort to capture all cases of maternal syphilis, in 2017, the Ministry of Health defined a case by any positive treponemal or non-treponemal test, regardless of confirmation, in pre-natal care and delivery. In an effort to standardize the case definition across the study years, we defined a maternal syphilis case as a reactive non-treponemal test with a reactive treponemal test. While the increase in 2017 and 2018 must be interpreted with caution, the rising maternal and congenital syphilis cases over the past decade suggest that this change in reporting likely did not substantially impact the linear trend. Third, Euclidean distance hot spot analysis may underestimate variance with <30 areal units of analysis, leading to a Type I error [26]. We cannot link HIV and syphilis databases to examine rates of co-infection, which is a potentially rich area for future research.”

Discussion:

This is the section where the authors should take advantage of data. Brazil is a continental country that presents a lot of diversity. The authors suggest managing syphilis as a separate pandemic, what it is, but it should take advantage of the organized care provided for HIV to implementing its actions. Although prenatal care is easy to access in Brazil and high access rates have been described, the great challenge is the quality of the offered care. It is common to offer rapid test for HIV in prenatal care and offer VDRL test for syphilis (even if the rapid test for syphilis is available). The authors should discuss these points to give us a better idea of the whole picture. I missed the issues related to HIV. There are many questions that could have been explored to compare the medical care of the two infections in the country.

Thank you for this point. We agree that Brazil is a diverse country, with one of the most successfully HIV care programs in the world. We also agree that access to prenatal care is not as much of a challenge as the quality, which has been raised by others as well. We have included another paragraph in the Discussion and modified the 2nd paragraph to address these issues: 

“The Unified Health System (Sistema Único de Saúde, SUS) HIV treatment program in Brazil is potentially one of the most comprehensive HIV public health programs worldwide, having been tremendously successful over the years [28]. The stable annual HIV MTCT rate over the past decade serves as a testament to the effective implementation of Option B+ guidelines, and universal ART coverage for all PLWH. In addition, the rise of raltegravir-based ART regimens may contribute to sustained viral load suppression among pregnant women, and prevention of HIV MTCT as a result in Brazil [29]. While efforts to curtail the maternal and congenital syphilis epidemics in Brazil have leveraged the existing HIV healthcare infrastructure, Brazil boasts a diverse population of over 200 million, and may require more tailored, region-specific strategies. 

 The increasing rates of maternal syphilis over the past decade are likely multi-factorial. Increased syphilis testing and enhanced surveillance as part of the Rede Cegonha and Projeto Nascer campaigns, public policy initiatives aimed at improving prenatal care in Brazil, are possible explanations for increased identification of maternal and congenital syphilis cases throughout the country [15]. While prenatal care is accessible to all pregnant women under the SUS network in Brazil, the quality may be suboptimal, particularly among women of a lower socioeconomic status [30]. In addition, other studies using spatiotemporal analyses have documented comparable increases in Brazil that cannot be explained by increased testing capacity alone [31, 32]. Inadequate treatment of partners, leading to re-infection of their pregnant partners, and lack of recognition of low-level titer infections (<1:16) without appropriate treatment despite the guidelines [33], have also been cited as possible explanations for rising maternal syphilis rates despite engagement in pre-natal care [3]. A recent study by Swayze et al. found that in a subset of women who fulfilled criteria for a syphilis diagnosis but did not receive penicillin treatment during pregnancy, 83% had been engaged in pre-natal care. This finding challenges the popular narrative that absence of pre-natal care fuels the maternal and congenital syphilis epidemics [3]. This may place unfounded blame on pregnant women and raises questions about this proposed causal relationship.”

In addition to the study with indigenous people mentioned in the discussion, there are other important references that discuss the situation of rapid tests in Brazil. The indigenous population is very peculiar and does not represent the national context. Authors should include other data on it.

 We thank the reviewer for this insightful comment. We have included other references re: the use of rapid syphilis tests in prenatal care. Respectfully, we would like to include this reference to the indigenous population, as they reflect one of the most vulnerable populations in the country. 

Reviewer #2: The manuscript aimed at spatiotemporal analysis and evaluation of hotspots of annual rates of detection of HIV and syphilis in Brazil from 2010 to 2018. The manuscript is well written and the study's outline is well outlined. However, authors should note a few points before publication

a) Exchange the expression CD4 for CD4 + T lymphocytes.

 Thank you for this suggestion. We have replaced “CD4” with “CD4+ T lymphocyte count” throughout the manuscript 

b) Were there any exclusion criteria in the methodology? For example, notification forms with no information?

 We apologize that we did not make this clearer. All cases of maternal HIV and syphilis must be reported to SINAN by law. Therefore, all cases of maternal HIV and syphilis reported to SINAN from 2010 to 2018 were included in our dataset. Missing data was treated as missing at random, and reported in Table 1 if applicable. The methods now delineate how we approached the dataset: 

“Data were provided by the Sistema de Informação de Agravos de Notificação (SINAN), the national reporting system for notifiable diseases of the Brazilian Ministry of Health, for all pregnant women with HIV or syphilis detected between January 1, 2010 to December 31, 2018 [17-19]. All cases of maternal HIV and syphilis are reported to SINAN by law. Data are publicly available if requested, although we used a combination of aggregate and individual-level data provided by the Ministry of Health through the Sistema Eletrônico de Informações ao Cidadão (e-SIC) [20]. The annual rate of detection of maternal HIV (per 1,000 live births) was defined by the Ministry of Health as a new diagnosis during pregnancy (two rapid tests or an ELISA with immunofluorescence/Western Blot), or an established HIV diagnosis. All cases of a reactive treponemal or non-treponemal syphilis test detected during pregnancy are reported to the Ministry of Health. In order to standardize the case definition across the years, and minimize the cases of false-positives or previous infections, we defined the annual rate of detection of maternal syphilis (per 1,000 live births) as a reactive non-treponemal titer (VDRL or RPR) with a confirmatory Treponemal test (fluorescent Treponemal pallidum antibody-absorption, micro-hemagglutination Treponema pallidum assay, ELISA, or lateral flow) [18] . . . Missing data were treated as missing at random, and reported in Table 1 if applicable.” 

c) In the legend of table 1, put the meaning of N/A.

 We have updated the Table 1 legend with the definition of N/A. 

d) Authors should include in the discussion a topic commenting on the possible causes of stability in the diagnosis of HIV infection.

 Thank you for this suggestion. This point was raised by multiple reviewers. We have updated the Discussion as such: 

 “The Unified Health System (Sistema Único de Saúde, SUS) HIV treatment program in Brazil is potentially one of the most comprehensive HIV public health programs worldwide, having been tremendously successful over the years [28]. The stable annual HIV MTCT rate over the past decade serves as a testament to the effective implementation of Option B+ guidelines, and universal ART coverage for all PLWH. In addition, the rise of raltegravir-based ART regimens may contribute to sustained viral load suppression among pregnant women, and prevention of HIV MTCT as a result in Brazil [29]. While efforts to curtail the maternal and congenital syphilis epidemics in Brazil have leveraged the existing HIV healthcare infrastructure, Brazil boasts a diverse population of over 200 million, and may require more tailored, region-specific strategies.”

Reviewer #3: This is a very well-written manuscript describing a compelling analysis of maternal HIV and syphilis epidemics in Brazil from 2010 to 2018. The authors found these epidemics to have some geographic distinction, and thus their conclusion—that the maternal syphilis epidemic requires a new, individualized approach to management—seems logical, appropriate, and important.

I have a few questions/comments:

1. Could the authors please further explain the process for detection of maternal syphilis? Does the definition (reactive treponemal + reactive non-treponemal) differentiate incident or prevalent infection from old, previously treated infection? The authors state that the Ministry of Health reviews each case, but what is included in this review (appropriate diagnosis? treatment? etc.)?

We apologize that this was not clearly explained in the Methods. We have updated the Methods as follows: 

“All cases of a reactive treponemal or non-treponemal syphilis test detected during pregnancy are reported to the Ministry of Health. In order to standardize the case definition across the years, and minimize the cases of false-positives or previous infections, we defined the annual rate of detection of maternal syphilis (per 1,000 live births) as a reactive non-treponemal titer (VDRL or RPR) with a confirmatory Treponemal test (fluorescent Treponemal pallidum antibody-absorption, micro-hemagglutination Treponema pallidum assay, ELISA, or lateral flow) [18]. Each case is reviewed by the Ministry of Health to ensure appropriate diagnosis, adequate maternal, partner and newborn treatment (with a penicillin-based regimen) and compared to previous infections if registered in the system to distinguish between past and present infections.”

2. Lines 112-113: The phrase “low non-treponemal titer infections” is unclear; consider rewording as “lack of recognition of low-level titer infections.” Additionally, the study cited here reports that these low-level titers are associated with lower likelihood of penicillin treatment but does not report that they are an explanation for rising syphilis rates. Perhaps the reference citation for this sentence is incorrect; I’m not sure that inadequate treatment of partners as a cause of rising maternal syphilis infections is discussed in this reference at all.

Thank you for this comment. We have replaced “low non-treponemal titer infections” with “lack of recognition of low-level titer infections.” Our team authored the paper cited, and argue that failure to recognize and appropriately treat low-level titer infections (<1:16) despite a clear indication by the guidelines, and absence of partner treatment, may be contributing re-infection and persistence of syphilis within sexual networks. We apologize that this was not clear, and have since re-rephrased the paragraph as such: 

“Inadequate treatment of partners, leading to re-infection of their pregnant partners, and lack of recognition of low-level titer infections (<1:16) without appropriate treatment despite the guidelines [33], have also been cited as possible explanations for rising maternal syphilis rates despite engagement in pre-natal care [3]. A recent study by Swayze et al. found that in a subset of women who fulfilled criteria for a syphilis diagnosis but did not receive penicillin treatment during pregnancy, 83% had been engaged in pre-natal care. This finding challenges the popular narrative that absence of pre-natal care fuels the maternal and congenital syphilis epidemics [3]. This may place unfounded blame on pregnant women and raises questions about this proposed causal relationship.” 

3. Please double-check if the percentage is 92 (rather than 83%) for cases of maternal syphilis in this study who were engaged in prenatal care.

We apologize about this confusion. We meant to refer to the 83% of women diagnosed with syphilis during pregnancy who did not receive penicillin despite engagement in pre-natal care. We have updated the paragraph as such: 

“A recent study by Swayze et al. found that in a subset of women who fulfilled criteria for a syphilis diagnosis but did not receive penicillin treatment during pregnancy, 83% had been engaged in pre-natal care. This finding challenges the popular narrative that absence of pre-natal care fuels the maternal and congenital syphilis epidemics [3]. This may place unfounded blame on pregnant women and raises questions about this proposed causal relationship.”

One additional minor comment:

1. Data is/was should be changed to are/were throughout.

Thank you for bringing this to our attention. We have changed this throughout the manuscript as requested. Thank you for your thoughtful comments.

---

## [Editor Report · Decision Letter 1]

21 Jul 2021

Maternal HIV and Syphilis are Not Syndemic in Brazil: Hot Spot Analysis of the Two Epidemics

PONE-D-21-12741R1

Dear Dr. Cambou,

We’re pleased to inform you that your manuscript has been judged scientifically suitable for publication and will be formally accepted for publication once it meets all outstanding technical requirements.

Kind regards,

Jodie Dionne-Odom, MD

Academic Editor

PLOS ONE
---

## [Editor Report · Acceptance letter]

26 Jul 2021

PONE-D-21-12741R1 

Maternal HIV and syphilis are not syndemic in Brazil Hot spot analysis of the two epidemics 

Dear Dr. Cambou:

I'm pleased to inform you that your manuscript has been deemed suitable for publication in PLOS ONE. Congratulations! Your manuscript is now with our production department. 

Kind regards, 

on behalf of

Dr. Jodie Dionne-Odom 

Academic Editor

PLOS ONE